# Interactive Exploration of Genomic Conservation

Venkat Bandi*       Carl Gutwin†

Department of Computer Science
University of Saskatchewan

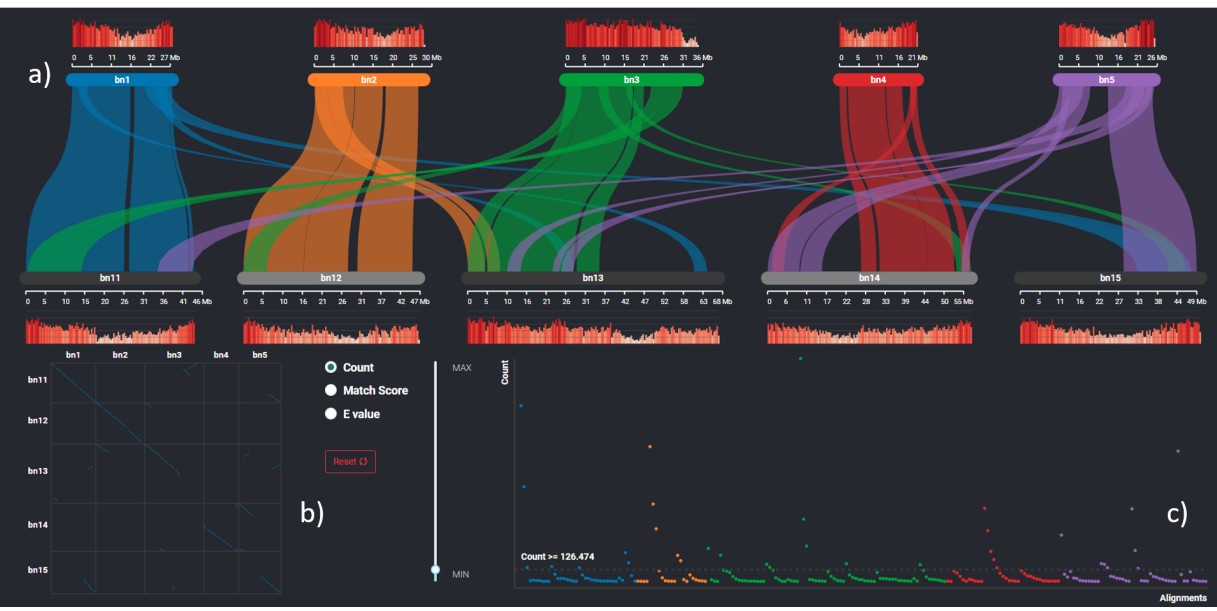

Figure 1: SynVisio visualization of conserved regions in the genome of Canola: a) Parallel plot with coloured ribbons linking regions of chromosomes where genes are conserved along with a corresponding histogram track showing gene density variation. b) Dot plot with chromosomes on the axes, showing every conserved gene as a point. c) Filter panel to refine the visualizations in real time with a graphical representation of the filtered parameter.

## ABSTRACT

Comparative analysis in genomics involves comparing two or more genomes to identify conserved genetic information. These duplicated regions can indicate shared ancestry and can shed light on an organism's internal functions and evolutionary history. Due to rapid advances in sequencing technology, high-resolution genome data is now available for a wide range of species, and comparative analysis of this data can provide insights that can be applied in medicine, plant breeding, and many other areas. Comparative genomics is a strongly interactive task, and visualizing the location, size, and orientation of conserved regions can assist researchers by supporting critical activities of interpretation and judgment. However, visualization tools for the analysis of conserved regions have not kept pace with the increasing availability of genomic information and the new ways in which this data is being used by biological researchers. To address this gap, we gathered requirements for interactive exploration from three groups of expert genomic scientists, and developed a web-based tool called SynVisio with novel interaction techniques and visual representations to meet those needs. Our tool supports multi-resolution analysis, provides interactive filtering as researchers move deeper into the genome, supports revisitation to specific interface configurations, and enables loosely-coupled collaboration over

*e-mail: venkat.bandi@usask.ca
†e-mail:carl.gutwin@usask.ca

the genomic data. An evaluation of the system with five researchers from three expert groups provides evidence about the success of our system's novel techniques for supporting interactive exploration of genomic conservation.

**Index Terms:** Human-centered computing—Visualization—Visualization systems and tools—Visualization toolkits; Human-centered computing—Interaction design—Interaction design process and methods—User Interface design.

## 1 INTRODUCTION

Genomic data is being generated at an unprecedented rate due to the emergence of new sequencing systems. The increased availability of high-resolution genomic data has aided researchers in tackling a range of questions spanning evolutionary biology, plant breeding, and medical research. One area where genomic data is extensively used is comparative genomics, which involves comparing genetic sequences between or within species to study their similarity. Sequence similarity often implies similarity of function, and can also shed light on the evolutionary relationship between sequences because as organisms evolve and diversify into different species, they retain parts of their DNA from common ancestors. The study of these conserved sequences within chromosomes of the same or different species is called **synteny analysis**. Researchers study these sequences by grouping them into contiguous blocks referred to as syntenic blocks and then analyse their properties such as location, size, and orientation.

Some aspects of large-scale genomic comparison are purely computational and thus can be automated, but human judgment is still

vital in comparative analysis and visualization tools can assist researchers in these tasks. The increasing size and complexity of genome sequences mean that the work that genomic scientists do with their datasets is constantly evolving: genome visualization tools are now used in diverse tasks such as evolutionary investigations of gene duplication events [35], missions to look for new medical treatments [2], and comparisons of gene expression to relate genotype and phenotype [11]. These kinds of complex tasks mean that researchers need access to systems that can support a wide variety of exploration, interaction, and collaboration activities – and the increasing need for interactivity coupled with the easy availability of datasets (e.g., through public databases such as NCBI and Ensembl) has led to a surge in the demand for computer-based support tools. However, current tools for visualizing and exploring genomic datasets have not kept pace with this increasing demand and are limited in their capabilities: they typically support only a small variety of datasets; they are not designed for investigation of complex synteny scenarios such as polyploidy (whole-genome duplication, which is common in plants); and they often do not support visualizations at multiple genomic scales. One reason for these limitations is that genomic visualization tools are rarely developed in close collaboration with the genomic scientists who actually use those tools, and as a result they do not consider the kinds of genomic exploration and analysis tasks that are now performed. For example, a task such as tracing the conservation of genes across more than one species requires the ability to explore pairwise comparisons at multiple levels; similarly, refining sequence assemblies requires annotating existing visualizations with gene density plots to verify assembly quality.

To address these limitations, we have been working with three teams of genomic scientists to understand the interactive and visual requirements for current genomic investigations. The three teams all study plants, but perform very different kinds of exploration and analysis. In collaboration with these experts, we identified six requirements for interactive genomic visualizations that are not supported by current synteny visualization tools: the need to refine datasets in real-time, the need to work with multiple perspectives on the data, the need for dynamic multi-resolution visualizations, the need to link secondary datasets to the genomic data, the need for new visualization of synteny across multiple genomes, and the need to support navigation and revisitation in genomic data spaces.

Based on these requirements, we designed a tool called SynVisio that has multi-scale visualization and interaction capabilities to meet the needs of genomics experts. SynVisio is a free web application [1] that visualizes the results of several synteny-detection systems. It lets researchers explore syntenic blocks through coordinated multiple views including parallel plots at several scales (Figure 1(a)), dot plots (Figure 1(b)), and a dynamic filter panel (Figure 1(c)) where users can refine the display of conserved regions based on similarity and the number of contiguous genes in a conserved block. Our tool enables further interactive exploration by letting users explore datasets at multiple resolutions (genome, chromosome, or gene), and by allowing users to annotate the genome with supplementary genomic tracks (such as gene density and copy number variation) using heatmaps, histograms (Figure 1(a)) or scatterplots. SynVisio also offers multi-level visualizations such as stacked parallel plots and hive plots to trace conservation across multiple genomes.

SynVisio has been extensively used by our three collaborating teams for more than a year (as well as by numerous other researchers around the world). We carried out a continuous real-world evaluation of our tool to assess its success in meeting the visualization and interaction requirements of expert users. The evaluation provides new evidence about the value of our innovations, such as using multiple views as overviews and providing adaptations of existing parallel plots for visualizations of synteny across multiple genomes.

[1]https://synvisio.github.io

Our experiences with the visualization system and its evaluation provide several contributions that can be valuable for other genomic visualizations: the identification of visualization and interaction requirements for real-world comparative genome analysis research; the investigation of how multi-scale and multi-perspective views are used in real-world analysis work; the design and evaluation of multi-level visualizations for displaying multi-genome conservation; and the design and evaluation of techniques to support navigation and revisitation in genomic datasets.

## 2 RELATED WORK

This research builds on three main areas of existing work: comparative genome browsers; interaction techniques used in genomic visualizations; and support for revisitation and tracking interaction history.

### 2.1 Comparative Genome Browsers

Regardless of the data type or domain, comparison is a common task in data analysis and visualization whenever there is a need to understand the relationship between a given set of items [8]. Visual comparison has been shown to improve the understanding of data in several kinds of visual presentations [42] since the earliest days of visualization, such as Playfair's use of line graphs to demonstrate the change in stock prices in relation to wars [3]. In the field of genomics, comparisons can aid biologists in a diverse set of tasks such as identifying functional elements, studying large scale rearrangements and genome evolution, and refining results of genome assembly systems using reference genomes [22]. Systems have been built to support visual comparison tasks at several different genomic scales from the nucleotide level all the way up to the whole genome level. The earliest examples for representing synteny involve the adaptation of dot plots used for comparing local alignments for larger sequences. Most of these tools primarily perform the actual genome level comparison and present their results through dot plots for closer inspection (e.g., DAGChainer [10] or the VISTA plots of the MUMmer alignment tool [16]). Dot plots are two-dimensional representations where genomes of two organisms are presented along the $x$ and $y$ axes. Such matrix-based plots offer effective genome-level summaries of alignments, but cannot be extended for multi-way comparisons between several genomes or used to identify smaller rearrangements at the chromosome level.

Another representation of synteny that has been used in tools such as Cinteny [39], Sybil [4], and MEDEA [24] is a pill-shaped ideogram at the chromosome level. In this design, chromosomes of the source genome are represented as pill-shaped blocks that are colour coded on a categorical scale; chromosomes in the target block are represented as similar pills with varying sizes based on their genomic lengths. Syntenic regions are then represented through colour-coded ribbons, where the colour is determined by the source chromosome. A problem with this representation and other designs that rely extensively on colour is that it cannot be extended for a large number of chromosomes as humans cannot easily distinguish beyond approximately ten colours [42]. A third form of representation that has recently become popular is the chord diagram (Circos-style) plot [15]. In these plots, genomes are represented as arcs in a circular layout, and syntenic regions are shown as lines connected through the middle of the circle. Although circular layouts are compact and visually impressive, their non-linear representation can cause visual cluttering and occlusion when exploring large datasets, emphasising the need for comparative tools that are built to support in-depth analysis rather than focusing only on presentation aesthetics.

### 2.2 Interactive Genomic Visualization Systems

Although static visualizations can assist users in simple data analysis, the static approach falls short for complex tasks and activities [5, 28, 29, 36, 41, 48]. A visualization's effectiveness can be

considerably improved by providing interaction mechanisms that modify the graphical representations based on the dynamics of different tasks, users, expertise, and context [36]. In comparative genomic visualizations, an early examples of adding interaction to basic charts is Dagchainer's ability to zoom into a two-dimensional dot plot for closer inspection [10]. Tools like SynMap2 [13] also allow exploration of dot plots through mouse-based scrolling and panning, similar to mapping platforms such as Google Maps. Zooming for exploration is also used in several other tools like Mizbee [21] and AccuSyn [23]. Mizbee presents overview-level information in a Circos-style plot, and specific sections of the genome can be zoomed and filtered through markers on the overview plot.

mGSV(Multi Genome Synteny Viewer) is another tool that provides a summary view of conservation at the genome level in a Circos-style plot and operates in both a pairwise view mode and multiple view mode [32]. In the pairwise mode users select specific regions of the different genomes and reorder the genomes, and in the multiple view mode users can toggle the visibility of genomic regions to avoid overlapping of genomes in a stacked layout. The mGSV browser also employs a heuristic algorithm to optimize the layout of the genome order based on the size of the conserved regions (to minimize visual clutter). However, this can often create layouts that, while being visually clear, may not provide the right biological context. The AccuSyn system addresses this problem through a novel human-in-the-loop method [23]; it uses a simulated annealing heuristic to arrive at the optimal layout and also takes into consideration the position of chromosomes set by the users through manual dragging and flipping operations. This can ensure that as users explore the syntenic relationship between the genomes, they can tune the algorithm to arrive at an uncluttered layout that also maintains meaningful context.

## 2.3 Interaction History and Revisitation Support

Exploring genomic information at different resolutions can be problematic due to a dataset's size and complexity, and due to limitations in the availability of visual space, but these problems can often be addressed through effective interaction techniques. However, this puts an additional burden on users to remember their location both in genomic space and in the parameter space of the visualization tool. If users cannot remember the positions of objects and markers in a visualization (or how they got there), it can be difficult to revisit them. Revisitation is important in data exploration as it can help users retrace their steps and is part of the *history* stage of Shneiderman's information-seeking mantra [38]. Revisitation can be compromised by interaction techniques that switch between graphical representations or zoom into a particular region – causing them to lose the context of their previous location. Humans are good at remembering object locations in visual workspaces, but visual context switches can disrupt this ability [34], and techniques such as fisheye views that distort the original visualizations can also impair spatial memory [40].

One method of addressing this problem is to store the interaction history of the system and present it as a graphical abstraction either in the visual system itself or in an external panel. Examples of direct interaction history encoded in a visualization system are a variety of "edit wear and read wear" techniques such as the "visitwear" mechanism that adds visual traces to visited nodes in a fisheye view [40]. *HindSight* is a similar direct encoding design framework; interaction history is encoded by making visited charts in a multi-plot system appear darker and by making visited lines in a chart slightly larger. Studies of this framework showed that users were able to visit more data points and recall novel insights better than with standard presentations [6]. A second indirect encoding method for interaction history stems from research in visual analytics for supporting *provenance* – the history of steps that led to a particular result in a data analysis workflow [7, 9]. Visual provenance has been

explored in systems such as the *VisTrails* tool [1] that let users save visual outputs and revisit earlier states in the data analysis process. In the context of genomic visualizations, to the best of our knowledge, direct encoding of interaction history has not yet been explored.

## 3 APPLICATION DOMAIN

The visualization system described in this paper was developed for the exploration of conservation in genomes. Our goal was to understand the scope of visualization in analysing genomic conservation and explore ways to improve this analysis process through novel interaction techniques and visual representations. In the following sections we first provide the biological background on genomic conservation followed by a characterization of genomics datasets.

## 3.1 Biological Background

Genomics is a field of biology that involves the study of genomes of various organisms to understand their structure, function, and evolution [25]. A genome is the complete set of DNA transferred from one generation to the next through self-replication [31]. Although cellular error-checking mechanisms ensure that the DNA being replicated is identical to the original, mutations can occur; as these mutations accumulate over time, they lead to the divergence of species. Understanding how these changes may have occurred is an important part of comparative genomics and has large-scale implications in issues such as the role of genetic factors in human health [2].

Compartive genomics looks at two or more genomes sequences to identify conserved regions, which can imply links between phenotypic and genotypic properties, and can help biologists understand these characteristics. For example, researchers were able to compare gene expression data of several plant sequences (which have high gene duplication rates) with evolutionary conservation data to improve gene discovery [11]. Conservation can be inferred by studying the collinearity of several genes over long regions. These regions are called **synteny blocks** and are the primary focus of this paper. With the availability of fully sequenced genomes for several model species, analysis of syntenic blocks can reveal evolutionary adaptions and also improve the transfer of knowledge to non-model organisms that have not been fully mapped [50].

## 3.2 Data Characterization

Synteny data is generated by combining positional information of genes along a genome sequence with pairwise comparison results to construct chains of collinear gene pairs. It is computed either by grouping neighboring gene pairs (OrthoCluster [49]) or by utilizing dynamic programming to create chains of pairwise collinear blocks around anchor genes (MCScanX [43]). These collinear blocks are the synteny blocks used in our system. Every such block of collinear genes has a set of metadata including a similarity score (indicating the quality of match), the size of the block (i.e., the gene count), the position of the block in the chromosomes of every genome where it appears, the orientation of the block (forward or reverse), and a probabilistic value indicating the likelihood of a true match. Every block also has a secondary layer of information in the form of a list of individual gene pairs in that block and their match scores. This data, combined with the structural information of the genome (size and position of every gene in every chromosome) can be used to associate every region of a genome with its duplicated regions in the other genome (or within the same genome). This information can be organized at three scales: *Genome → Chromosome → Collinear Gene Block*. The data thus gives a structural map of every duplicated region in a genome and its sub-elements at multiple levels.

## 4 REQUIREMENTS FOR SYNTENY EXPLORATION

Visual synteny analysis focuses on showing conserved regions in order to provide information about their location, size, and orientation,

along with other attributes. While most existing synteny analysis tools allow for investigation of these basics, they do not support the diverse analysis tasks that can vary widely based on the underlying biological question and the genomic data under investigation. The activities in synteny analysis can also vary at different genomic resolutions: for example, the chromosomal location of large collinear blocks can assist researchers during genome assembly whereas the order of individual genes in a single collinear block can inform researchers about the function of these genes [12]. This means that the design of a comparative visualization needs to be carried out in the context of real-world datasets.

To gather real-world design requirements, we have been working closely with three teams of genomic researchers working in different areas of biology, plant science, and crop breeding. Over the past three years, we conducted numerous interviews with these groups to understand their genomics tasks and discover and refine interaction requirements. The sessions broadly revolved around understanding the kinds of tasks researchers perform when analyzing genomic conservation and looking at the shortcomings of the existing tools. Although all three groups are primarily interested in analyzing synteny in plants, their individual use cases and datasets varied widely, providing us with a diverse set of user scenarios.

The first research group investigates genomic conservation in the *brassica* genus, as it offers an ideal model to study polyploid evolution, which is responsible for genetic variations that are advantageous from an evolutionary perspective [17, 19]. Synteny analysis is used by this group to assess the quality of their assemblies of the canola genome (which is allotetraploid, meaning that it has four copies of every gene). The second research team looks at genomic conservation in pulses (e.g., lentils and chickpea) to improve various agronomic traits, as these are important and widely grown legume crops. The requirements here were largely focused on cross synteny (between species) rather than self synteny (same species). Researchers in this group emphasized the need for an adaptive analysis system based on the genome size due to issues with existing tools that do not allow adequate comparison of datasets with large difference in their genome sizes (for example, the chickpea genome is 740 Mbases and the lentil genome is 4 Gbases). The third research group investigates genomic conservation between the three subgenomes of the hexaploid bread wheat genome, and thus wanted a system that could visualize multi-way synteny between the three sub-genomes instead of the standard two-way analysis (i.e., single source genome and single target genome) that current tools offer. Our discussions with the three research teams led to the following six major requirements for interactive genomic visualization systems:

$R_1$. **Dynamic refinement of visualizations**. Synteny analysis focuses on identifying conserved regions in specific parts of the genome with the ability to focus on distant or close matches from an evolutionary perspective. Researchers need to be able to filter the generated visualizations in real time based on features of the conserved region such as the level of match and its chromosomal position.

$R_2$. **Multiple perspectives on the dataset**. Researchers who undertake complex analysis tasks require multiple coordinated views that show different visual representations, each focusing on a particular primary feature of the dataset such as the orientation (dot plot) or the location (parallel plot). Researchers integrate the different perspectives in different ways as they carry out their tasks.

$R_3$. **Dynamic visualizations of multi-scale data**. Genetic conservation is often explored at several levels (genome, chromosome, or gene block), and the focus of analysis can be different at every genomic scale. Visualization systems should therefore

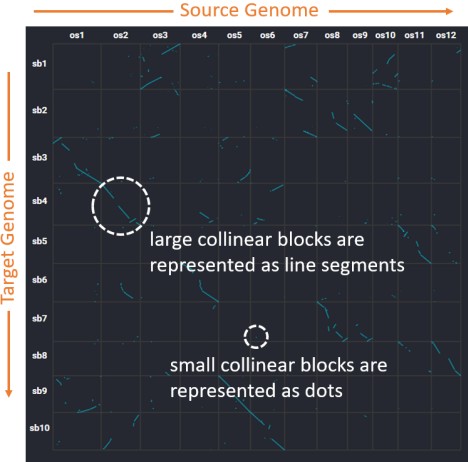

Figure 2: Encoding of conservation at the genome level in a dot plot.

allow users to switch scale quickly and easily, and should provide capabilities and interactions that adapt based on the scale of the investigation.

$R_4$. **Augmenting visualizations with secondary data**. Insights in synteny analysis can be gained by looking at conservation in the context of gene density or the positions of genetic anomalies (single nucleotide polymorphisms). Therefore, systems should offer researchers the ability to add layers of visual information onto the basic visualizations, using the main representation as a reference frame.

$R_5$. **Visualizations of multiple genomes**. Multi-way visualizations can let researcher trace conservation across several genomes, thus offering a novel way to visualize synteny combined with phylogeny (i.e., the evolutionary relationship between two species).

$R_6$. **Navigation and revisitation support**. Synteny analysis is often used in the hypothesis-generation stage of research, requiring that genome scientists explore several scenarios through the analysis tool. This can be problematic when genomes are large (e.g., the wheat genome is 17 Gbases – six times larger than the human genome), because researchers can easily lose context of their location in the genome (particularly in polyploid organisms where genes are duplicated multiple times). In addition, a complex visualization system also presents a large "parameter space" requiring that users remember the settings and navigation actions that brought them to their current viewpoint. Analysis tools therefore need to support navigation and record provenance in order to enable communication between collaborators and to enable revisitation of potentially-interesting locations during exploration.

## 5 VISUAL ENCODING OF CONSERVATION

Visualizing synteny is a multi-faceted problem because both the visual representation and the resolution of the information can change based on the underlying biological question. In designing a solution for this problem, we started with the taxonomy of design space created by previous synteny visualizers like Mizbee [21] which identified the 4 major features of syntenic data that are used in its analysis: proximity/location, size, orientation and extent of similarity. Mizbee visually encoded these features through linked connections between two parallel segments and further relies on colour to encode additional features. We extend this design space by providing a

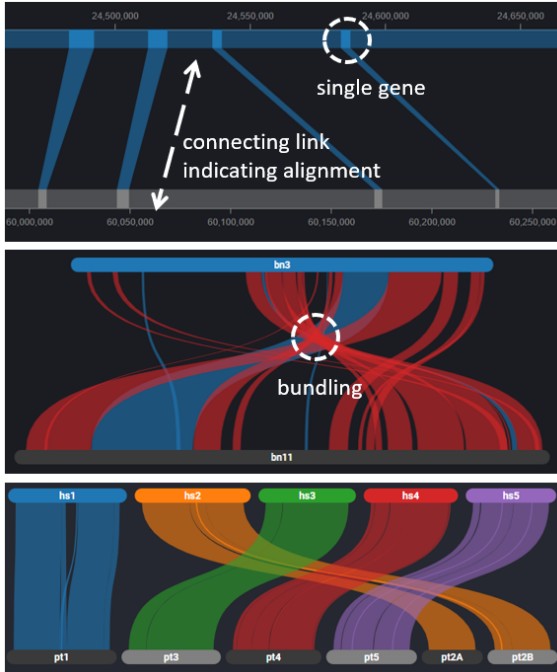

Figure 3: Encoding of conservation in parallel plots at different genomic scales. Top: gene block level with parallel tracks showing gene coordinates. Middle: chromosome level where orientation is encoded through color (red for inverted blocks, blue for forward blocks). Bottom: genome level, where color encoding is based on the source chromosome.

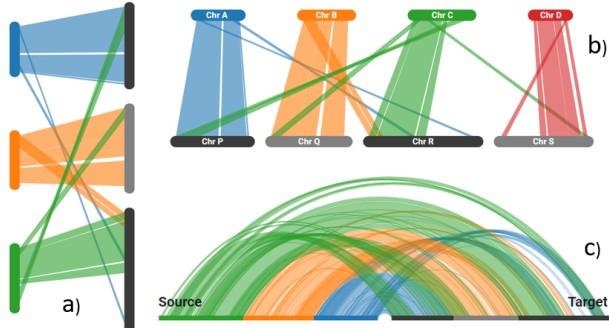

Figure 4: Different layout strategies for the parallel plot where conservation is encoded through connecting ribbons : a) Vertical Layout, b) Horizontal Layout, and c) Adjacent Radial Layout.

foundation for novel interactive features to refine the datasets and explore them across different genomic scales. In designing SynVisio we used parallel plots and dot plots as primary forms of visual representation, and adapted these to create multi-level hybrid plots to represent synteny in multi-way comparisons.

## 5.1 Dot Plot

Any encoding of conservation in genomes maps the location, size, and orientation of conserved regions over specific genomic coordinates. Dot plots (also known as scatterplot matrices) achieve this mapping through the positional encoding of similarity over a two-dimensional matrix. The source and target genomes are placed along the $x$ and $y$ axes, respectively, and collinear blocks are marked as either dots or lines based on their size, as shown in Figure 2. Gridlines are then added to the plot to indicate chromosomal boundaries. The visual encoding in this representation can be extended to other scales by changing the genomes along the $x$ and $y$ axes to view individual chromosomes or smaller gene blocks. The dot plot's matrix-based representation effectively provides an overview of the dataset and can be used to highlight breaks (shown as gaps in a line), inversions (shown as lines that are perpendicular to the diagonal), and duplications. However, dot plots are simplistic and can be difficult to understand, making them less suitable for exploration tasks.

## 5.2 Parallel Plot

For this representation, we adopt a design that represents synteny through a combination of positional encoding for genomic location and connections for similarity. In this approach which is based on sankey diagrams [42], two sequences are stacked horizontally parallel to each other, and conserved regions are connected through ribbons. Secondary encoding in the form of colour is used to show additional information about the conservation. At the gene block

level, every gene in the block is linked to its corresponding duplicate gene by a ribbon whose width is dependent on the size of the gene (Figure 3 (top)). The blocks are annotated with numeric tracks corresponding to their position in the chromosome, and color encoding is used to distinguish the source gene (blue) and target gene (grey).

At the chromosome level every connected ribbon in the parallel plot represents a single block of collinear genes; the width of the ribbon is based on the number of genes in the block. To show the orientation of the gene block, secondary encoding through colour is used to visually distinguish inversions, as shown in Figure 3 (middle) where forward matches are blue, and reverse matches are red. Unlike at the gene block level, at the chromosome level several bands can overlap and cross each other due to multiple gene translocation and inversion events, and this can lead to visual clutter. To mitigate this problem an edge bundling strategy is adopted: curved complex polygons are used instead of rectangles for the connecting ribbons. They are generated through **B**-spline curves [30] with control points set to bundle the curves towards the centre.

To further reduce visual clutter, connected ribbons are rendered with partial transparency, which ensures that even if ribbons overlap each other, they still remain visible. Finally, at the genome level, chromosomes are represented as rectangular pills whose width is relative to the size of the chromosome. Connecting ribbons are then used to link blocks of collinear genes in a similar manner to the previous level but are coloured based on the chromosomal source for every block, as shown in Figure 3 (bottom). This color encoding scheme is used because the origin of conserved regions takes priority over their orientation at the genome level, where researchers are mostly interested in identifying large-scale synteny patterns.

## 5.3 Layout Strategies

A common strategy used in all three parallel representations is the vertical separation between the source and the target to visually distinguish the two regions. This is easy to implement at the gene-block level and the chromosome level as the source and target regions are single continuous entities, but requires adaptations at the genome level. The genome is a combination of several chromosomes, so each chromosome has to be individually distinguishable while still being represented as a part of the whole source group and different from the target group. To achieve this grouping, we use the visual law of proximity from Gestalt principles [47], which states that proximity can override other visual similarities (shape, size, colour) to differentiate a group of objects. We represent each chromosome as a pill-shaped region and then lay them out end to end horizontally (Figure 4(b)) with small gaps between them to cluster the source and the target regions into two separate visual groups.

In exploring different layout strategies, we looked at several al-

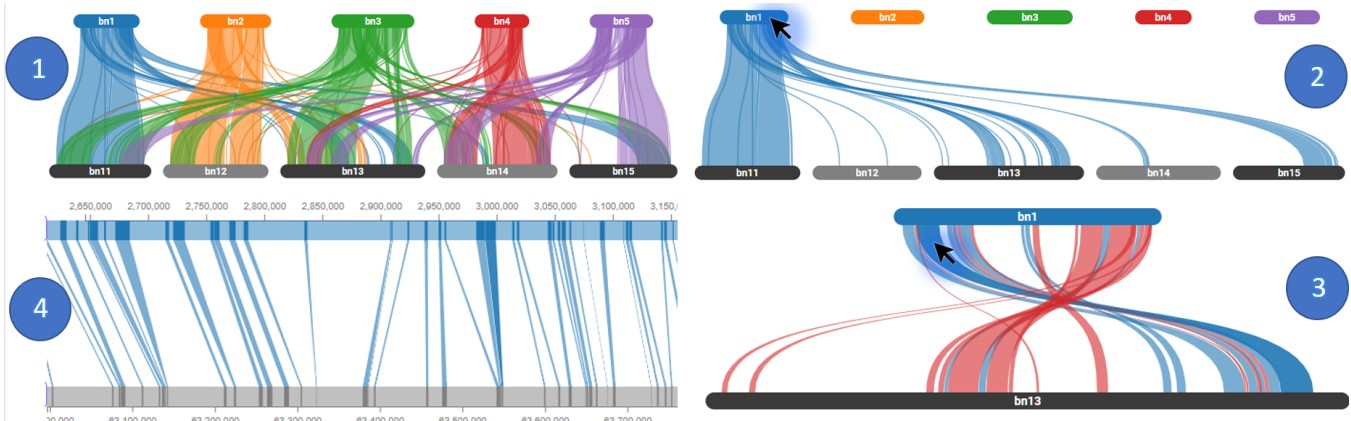

Figure 5: User interactions in exploring conserved regions in a top-down approach: four steps pictured in clockwise order.

ternative ways of arranging the chromosomes contiguously and in the order they appear in the genome. We restricted our exploration to linear layouts, as circular layouts are less suitable for exploratory analysis scenarios due to visual clutter and occlusion. In a linear layout, source and target chromosomes can be arranged either vertically (Figure 4 (a)), horizontally (Figure 4 (b)), or adjacent to each other in a radial layout (Figure 4 (c)). Our informal usability tests showed that of these three, the horizontal layout is the most useful, particularly given the width of many genomes and the aspect ratio of landscape-orientation monitors. The curvature of connected ribbons in the radial layout can interfere with the edge bundling strategy and cause unnecessary visual clutter, making the conservation harder to see and follow. However, the radial layout has advantages for multi-genome visualization when combined with bi-directional linking of radial nodes (discussed below).

## 6 SYSTEM DESCRIPTION

Here we outline SynVisio's interactive features and novel visualizations that address the six major requirement areas.

### 6.1 Composite Analysis Dashboard

The usefulness of a particular visual representation depends on the biological relationship under investigation, and this has created a need for an adaptive system that can be used in a wide range of scenarios. The inherent complexity in synteny data means that exploring it becomes challenging as the size of the dataset increases, and so instead of relying on a single visualization, information about conservation at every scale is presented through multiple coordinated views in a composite dashboard to meet requirement $R_2$. This is built on the underlying premise that users will have a better understanding of their data if they interact with it and view it through different representations [33]. Our design of multiple views that support the investigation of a single entity follows the guidelines set by earlier research into multiple coordinated views for information visualization systems [44].

The composite dashboard, as shown in Figure 1, has three distinct views, each highlighting a unique facet of the dataset : parallel plot, dot plot, and a filter panel consisting of a scatterplot with a control slider. The parallel plot offers position and location information about the conserved regions at a glance while the dot plot can easily highlight reversals and deletions within the conserved regions. Both views are linked to each other to ensure that users do not lose context of their interaction (this is done by mirroring the actions between the two plots). The filter panel consists of a control slider which can refine the data being visualized in the other two views dynamically, based on secondary features of the conserved regions such as the

match score and gene count. This slider controls the visibility of conserved regions in the other two views and can thus be used by researchers to look for evolutionarily distant or nearby matches. This feature combined with the two coordinated views offers researchers the ability to refine visualizations dynamically, thus meeting requirement $R_1$. Finally, a scatter plot is also provided in the filter panel that visualizes the secondary filter parameter, to assist researchers in making an informed decision during the data-filtering step.

### 6.2 Multiscale Exploration

To meet requirement $R_3$, SynVisio allows for exploration of conservation at different genomic resolutions spanning investigation from micro synteny all the way up to high-level genome duplication events. Information is presented in a top-down tiered approach in three distinct levels: the user starts with the entire genome, can then step down into an individual chromosome, and finally focus on the gene block level. Users can start their investigation at either the genome level or the chromosome level and interact with the visualizations in real time to look at conserved regions in a particular chromosome or drill down into the dataset all the way down to an individual gene in a conserved block, as shown in Figure 5. Additional details about the conserved regions are shown in a tooltip upon mouse hover, in either of the two views. At every level visualizations use scalable vector graphics and support zooming through mouse scrolling for closer inspection. Finally, SynVisio also offers users the option to flip the source or target at every level to investigate inversions in the syntenic blocks.

### 6.3 Support for Tracks

The ability to annotate visualizations with secondary datasets in the form of *tracks* can help researchers better understand the data under investigation, because information such as gene density and SNP (single nucleotide polymorphism) variations can highlight regions of interest in the entire genome. To meet requirement $R_4$, our tool offers researchers the ability to visualize secondary tracks parallel to the primary visualizations. Researchers can toggle visibility of the tracks through an on-screen button that is automatically enabled when the system detects track data in the input dataset. SynVisio offers four types of track visualizations: heatmap, histogram, linechart and scatterplot. Five equidistant grid lines are also provided in all the tracks as background reference scales. For the parallel plot, the tracks are added on the outer side of the visualization with one track above the source genome and the other track below the target genome (Figure 1 (a)) and for the dot plot, the tracks are placed beside the $x, y$ axes.

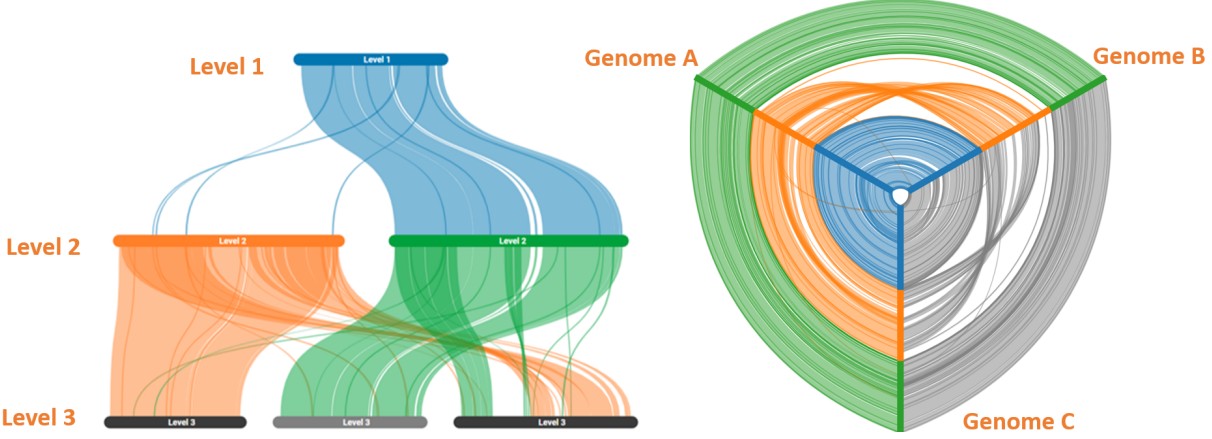

Figure 6: Multi-Genome Visualizations: Tree plot (left) and Hive plot (right).

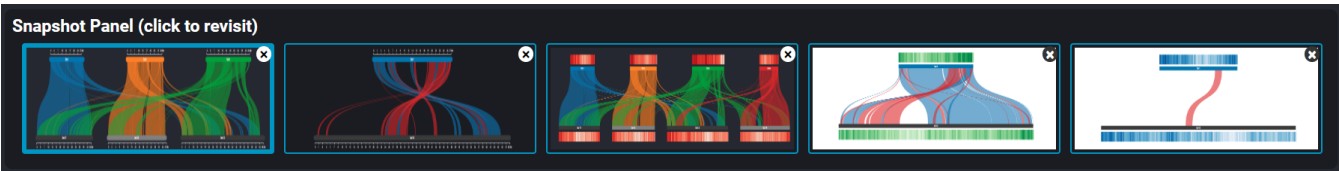

Figure 7: Snapshot panel showing stored visual states as thumbnail images that can be revisited or shared.

## 6.4 Multi-Genome Analysis

To address requirement $R_5$, SynVisio offers two multi-level visu-
alizations – Tree plots and Hive plots – which provide researchers
the ability to explore conserved regions between several genomes
instead of a single source and target. Both the visualizations are
extensions of the primary parallel plot. The tree plot is generated
first by stacking several parallel plots horizontally one over the other.
Conserved regions in the chromosomes in every row are then linked
bidirectionally as shown in Figure 6 (left). Starting from the top
layer, every chromosome acts as a parent node and is linked to the
chromosomes in the row directly below it if conservation exists, thus
forming a tree-like pattern that can be used to trace conservation
across several evolutionary levels.

The second visualization is a hive plot that is an extension of
the radial parallel plot layout. It offers a high degree of perceptual
uniformity and is based on a linearized network layout where nodes
are placed in radially-oriented axes, and edges are drawn between
the nodes to encode additional information [14]. In our tool, the
nodes represent chromosomes, and they are ordered sequentially
based on their order in the genome; conserved regions are then linked
through connecting ribbons. Unlike pairwise comparison, hive plots
do not have a single source axis as all axes are uniform in a multi-
way comparison scenario. Therefore, the connected ribbons are not
coloured to represent the source chromosome but are instead shown
in translucent gray. User interactions with the hive plot are then used
to select the source axis: when a user clicks on a particular radial
axis, all the connected ribbons emerging from it are coloured based
on the chromosomes they belong to in that axis, as shown in Figure
6 (right - Genome A selected). This form of variable encoding based
on user choice can be useful in selectively highlighting patterns in
conserved regions in a single genome against the background of
overall conservation.

## 6.5 Snapshot Panel for Revisitation and Collaboration

SynVisio allows for data exploration by changing the visual repre-
sentation of the data at different genomic scales, different filtering

scenarios, and different resolutions as the user zooms in for a closer
inspection. Even though humans are good at leveraging spatial cog-
nition to remember locations of objects in information workspace
tasks [34], context switching between different viewpoints during
exploration can disrupt this ability. To address this issue ($R_6$) SynVi-
sio lets users keep track of their actions through a visual snapshot
feature. It preserves the sequence of actions that led to the current
state of the visual interface in a snapshot. Users can store this snap-
shot at any time by clicking a save button at the bottom left corner
of the interface. These snapshots are stored sequentially and are
available for revisitation through a *snapshot panel* – in this interface,
snapshots are shown as a series of clickable buttons consisting of
a thumbnail image representing the visualization at that point as
shown in figure 7. Clicking on any snapshot button in the panel au-
tomatically recreates the stored visual state of the system. Snapshots
are valuable for an individual user to get back to a previously-visited
location, but the snapshots can also be sent to other users (as a web
link) to bring a collaborator to the same view.

## 7 User Evaluation and Feedback

SynVisio has been operational for more than a year, and has been
extensively used by our three expert groups throughout that time (as
well as by several other users around the world). We have carried
out several evaluation activities during the year, including regular
meetings with the experts to discuss the system's functionality and
usability. At the end of the year, we also carried out interviews with
five domain experts from the three expert groups.

In addition, to provide evidence on the effectiveness of our system
"in the wild," we also explored user activity logs on the website over
the year.

The interview study was conducted using semi-structured inter-
views with five domain experts from our three collaborating research
teams. The interviews were conducted either by phone or in person,
and each lasted around 60 minutes. All domain experts had con-
siderable exposure to SynVisio and were familiar with the different
features provided by the system; they were asked to give comments

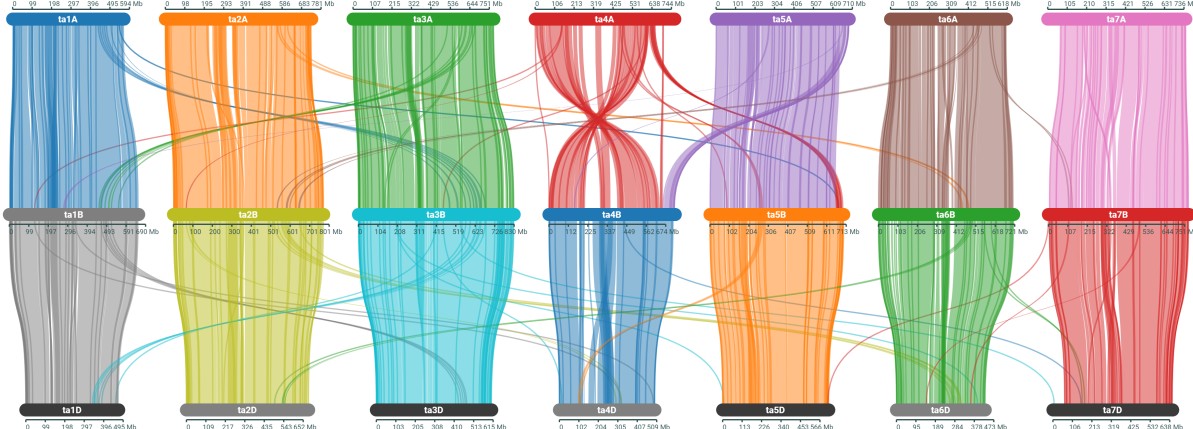

Figure 8: Collinearity between the three sub genomes of wheat A (top), B (middle) and D (bottom) presented through a multi-genome Tree plot.

on their use and assessment of the different features through open ended questions. The feedback from the user study is presented through the following two case studies showing the ability of the system to support diverse usage scenarios.

### 7.1 Case Studies

#### 7.1.1 Wheat *(Triticum aestivum)*

Wheat is one of the most widely cultivated crops in the world and plays an important role in human nutrition. It is a diverse crop that is capable of tolerating mutations and extensive hybridization. Wheat has a hexaploid genome (i.e., six copies of every gene) and is the result of a series of hybridization events between three ancestral genomes (A *(Triticum urartu)*, B *(Aegilops speltoides)* and D *(Aegilops tauschii)*) which makes it an interesting subject for synteny analysis. Our collaborators were involved in sequencing a high-quality reference version of the wheat genome and used our system to assess the quality and contiguity of the genome assembly through alignments between the sub genomes (A, B and D). As this analysis involved studying self synteny between multiple sub genomes, our collaborators relied on the multi-genome tree plot to summarize the large-scale chromosomal rearrangements and inversions between the three genomes as shown in figure 8. They found the stacked multi-level representation to be better than existing alternatives like Circos plots for tracing such conservation as there is a clear visual distinction between the three sub genomes - *"This tool is better than a Circos plot, especially when comparing multiple genomes, circos can be limited because you are seeing too many chromosomes in one circle and so are losing information ... a stacked layout like yours is easier to see." (R4)*.

Also synteny analysis of wheat is complicated due to the large size and complexity of its genome (17 Gbases – six times larger than the human genome); however, the multi-scale exploration features of our tool helped our collaborators in incrementally exploring the genome through tiered visualizations - *"...a single wheat chromosome is vast and wheat has 21 of those [chromosomes] placing stress on an analysis pipeline in terms of complexity...it is also very repetitive...so this [tool] is really neat. this is also very useful" (R1)*. Finally, in addition to their analysis activities, this group used the visualizations to present assembly results to their peers - *"the images have been used in presentations, academic meetings such as the international wheat congress and plant-animal genome conference [2019]." (R1)*.

#### 7.1.2 Canola *(Brassica napus)*

Canola is an important oilseed crop as it is an excellent source for both animal feed and high quality edible oil [37]. It is an allote-

traploid species (i.e., four copies of every gene) that was formed through interspecific hybridization of two diploid ancestors *Brassica rapa* and *Brassica oleracea* [26]. Studying this hybridization can help researchers in looking at genetic variations that are advantageous from an evolutionary perspective in polyploids. Our collaborators from this research group were interested in using comparative mapping to understand the level of genome duplication in modern brassica cultivars and the occurrence of genomic rearrangement in the evolution of these varieties from a common ancestor. They used our system to explore both self-synteny within canola and also cross-synteny between canola and its closely related relatives. The researchers in this group were particularly pleased with the composite dashboard in SynVisio, consisting of multiple coordinate views - *"I think it's quite good, I do really like that there's also the dot plot, in the corner, so that if anything is a little bit unclear, from the parallel view, you can kind of refer back to that." (R5)*. This comment highlights the value of multiple representations that provide a variety of perspectives during different types of analysis. This need for multiple representations for different use cases is further highlighted by comments made by some researchers who mentioned that in certain scenarios parallel plots were better - *"We have always used dot plots but these [parallel plots] are visually more intuitive...when chromosomes start breaking apart its much more difficult to follow where things are going in that big square...Its much easier to trace things and work out where you are...(R3)"*. Finally our system also helped researchers in this team at refining their assemblies proving its usability across different scenarios - *"Our assembly got better when we upgraded our sequencing from short read to long read sequencing technology as more regions are assembled. This tool helps us visualize that improvement ...(R2)"*.

### 7.2 Analysis of Web Traffic Logs

To quantify the use of SynVisio since it was made public we analysed web traffic through Google Analytics from January 2019 to the end of December 2019. In this 12-month period, SynVisio had 154 unique users spanning 267 sessions, with session times ranging from two to 28 minutes. Users were from 18 different countries with a majority from China (53) followed by the United States (45) and Canada (23). Although our system was designed solely based on requirements from our collaborators in our university its use by researchers across the world in a wide variety of research projects shows that it can support diverse datasets and user scenarios.

### 7.3 Evaluation Summary

Although several visualization tools exist for exploring genomic conservation, they are limited in their usability as mentioned by

the researchers we interviewed - *"There are a couple of R based tools that we use but none of them are as complete as SynVisio (R4)"*. SynVisio has thus been able to fill a critical gap by offering researchers a novel way to interact with their datasets in real time - *"There isn't anything like this. Especially not where you can play around with your dataset. (R3)"*. When asked to rate the usability of SynVisio as tool to explore genomic conservation, on a scale of 1 (very bad) to 5 (very good), four researchers gave the system the highest rating of 5/5 and one researcher gave it a rating of 4/5. The success of the system is also shown by the wide range of users from around the world. Further, images generated by SynVisio have also been used in several research publications describing new genome assemblies and annotations highlighting the popularity of our hybrid multi genome visualizations [20, 27]. Finally, our system design code has been open sourced on GitHub [2] under an MIT license and has already been adopted into several online genome databases, such as TeaBase (tea) [45], VitisGDB (grapevine) [46] and SilkDB 3.0 (silkworm) [18], proving its value for exploration of diverse genomic datasets.

## 8   DISCUSSION AND FUTURE WORK

The different interactive features in our system along with the hybrid multi-scale plots provide a generalized visualization framework for the design of other analysis tools that work with genomic data. Most visualization tools in this area work solely as chart generation systems for research publications. However our experience with SynVisio has shown that coupling real-time interaction with visualizations offers biologists new ways to explore their datasets, moving visualization systems into the hypothesis testing phase of a biologist's research process instead of the report generation stage.

In addition, genomics datasets are abstractions of complex interconnected biological systems and thus have some unique characteristics – such as being extremely large in scale with sparse distribution of patterns. These datasets are also spread over multiple scales and interactions can occur between distant sections with numerous overlaps. Our system tackles this complexity by visualizing these datasets through tiered plots that are adaptive to the scale of the datasets. We address the problem of overlaps between distant connections, to a limited degree, through an edge bundling strategy; however, there can still be some amount of visual clutter due to the natural ordering of the chromosomes in the layout. In future we will address this by optimizing the layout to minimize clutter and offering users greater control over the layout through drag-and-drop interactions on the chromosomes.

Finally, visualizing syteny in stacked parallel plots offers a novel way to trace conservation across several genomes, and the insights gained from this representation can be improved further by also encoding the direct evolutionary relationship between these genomes. Such relationships are commonly represented as phylogenetic trees where the pattern of branching in the tree is indicative of shared ancestry. Combining such branching techniques into the existing multi-level visualizations such as tree plots would offer researchers a novel way to analyse large scale datasets such as pangenomes (i.e., all genomes present in a particular strain of a species).

## 9   CONCLUSION

Comparative analysis of genomic conservation provides biologists crucial insights into an organism's evolutionary history and internal mechanisms. However, current tools for visualizing conservation lack the features needed by researchers to explore diverse biological scenarios. To address this issue, we identified six major requirements for interactive genomic visualization systems through collaboration with domain experts. Based on these requirements, we developed SynVisio, a web-based visualization tool for exploring genomic conservation through novel interaction and representational capabilities.

---

[2]github.com/kiranbandi/synvisio

SynVisio offers dynamic visualizations in coordinated views to explore genomic data at multiple genomic resolutions, and also offers hybrid visualizations to trace conservation across multiple genomes. A continuous real-world evaluation of the tool provided evidence of its effectiveness at tackling a wide range of genomic analysis tasks.

## ACKNOWLEDGMENTS

The authors wish to thank Isobel Parkin, Andrew Sharpe, Kirstin Bett, and Kevin Koh for the expert knowledge they brought to this project. This work was supported in part by the Natural Sciences and Engineering Research Council of Canada (NSERC), and additionally by the Plant Phenotyping and Imaging Research Cenre.

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
