# OpenReview forum: "Interactive Exploration of Genomic Conservation"
_graphicsinterface.org/Graphics_Interface/2020/Conference — GI 2020_

### Official Review · AnonReviewer1 · 2020-04-17
**Good work but need stronger case studies**

**Rating:** 6
**Confidence:** 4

**Review:**

This paper presents an interactive visualization system for exploring genomic conservation. The design is based on a few existing visualizations such as dot plot and parallel plot. The system was developed by involving domain experts and evaluated through deployment studies. Overall, it seems a strong paper. It is clearly written. The visual design looks appealing.

One issue I had with this paper is its contributions. It says that Sybteny has "novel visualizations such as stacked plots and hive plots," however, I didn't find sufficient novelty in its visualization design. These charts exist and have been widely used in many applications. A better story of this paper should focus on the design study itself, including the requirement gathering, design process, and deployment. While I think this system can be useful by putting together different views to serve the purpose, I don't think it is novel enough, at least not the selling point of this paper.

Focusing on the actual design study, I found several details are missing. I'd like to know the design process and how the requirements were gathered, via interviews, or focus groups? The paper just briefly mentions "our discussion with the three research teams led to..." But how?

Further, I was very confused about the so-called taxonomy of design space created by prior visualizations. It lists a few existing plots, but I don't think it can be called a design space. Design space is a multidimensional combination and interaction of factors under consideration. The content of this section can be integrated into the actual system description and added with justification for fulfilling the design requirements.

Another big concern is that the case studies lack depth. While presenting three case studies look good, none of them reveal interesting insights. The system has been up running for a year. There must be a lot of data to collect. I'm disappointed with the three toy examples, which lacks sufficient details in how the analysts use the system. I suggest removing one or two case studies and describing one in greater detail, through a step by step demonstration of the usage of the system with figures illustrating the insights found. As the system is deployed, some interaction log analysis is necessary, in addition to just the summary web traffic.

In summary, the evaluation part of this paper needs to be strengthened. This is because the visualization design is not that novel and I view whole design study as the core contributions.

---

### Official Review · AnonReviewer2 · 2020-04-19
**A visual-appealing system, lack detail of qualitative research**

**Rating:** 5
**Confidence:** 4

**Review:**

This submission described its visual analytics system for exploring genomic conservation data. I appreciate the detailed description of the proposed system. However, I think the system lacks novelty, and the evaluation was insufficient as evidence for its effectiveness. Meanwhile, I believe the qualitative research part should be a strength of the paper. However, the current description is not enough.

One of the most exciting parts of this submission to me is how the authors understand the domain expectations of a visual analytics system. However, the part is only briefly covered. For example, I would like to know how the authors discussed with the domain experts (e.g., interview, workshop, focus group), what data collected in these processes, and how the authors analyzed the collected data (coding, etc.).

Qualitative evaluation is considered as a standard way for domain-specific visualization systems. However, the same as the final evaluation interviews as to the requirement analysis, a scientific method to plan the evaluation, collect the data, and analysis are needed. Just mentioning the experts think your system is sound is not enough.

Some of the terminologies used by the authors are inconsistent with the visualization community. For example, the parallel plot and the tree plot from the authors are considered as sankey diagram in the visualization community. The dot plot is a scatterplot matrix. The authors also claimed the tree plot (sankey diagram) and the hive plot as the novel, but they are not. The proposed system basically is a multiple coordinated view system with existing visualization views and common interactions. I do not think there is too much novelty in the design, but I also understand this is not the focus of this submission.

I have a few comments for the proposed system as well:
 * The authors barely mentioned the histograms appear in the teaser figure.
 * The filter can only work on one property, is that sufficient?
 * From the video, the snapshot is presented as a labeled button, and I suspect anyone can recall useful information from a button like that. A thumbnail image may be more useful.
 * The figure of the Hive plot is not informative. It is too cluttered, while the one presented in the video seems to be a better example.

Overall, I think the authors are contributing to the field of genomic conservation by providing a good visual analytics tool. From the statements from the authors, the tool seems to be quite popular in the field with real users, which is quite amazing. However, I think the current presentation of the work is not ready for publication. I think the authors may reduce some descriptions of the system and add more detail of the qualitative research. I would suggest not accept it at its current stage.

---

### Official Review · AnonReviewer3 · 2020-04-20
**a great read and balance between application domain and visualization research**

**Rating:** 9
**Confidence:** 4

**Review:**

This design study paper describes a tool to interactively visualize conserved genomic regions. It describes the problem domain and provides a data characterization. The paper provides 6 requirements on which the tool is build on. The visualization system consists of multiple coordinated views including known plots such as Dot Plots and Parallel Coordinate plots, but also includes two novel visualizations Tree plot and Hive plot. The paper ends with a description of a user evaluation based on semi-structured interviews conducted with 5 domain experts of 3 different research groups the authors collaborated with.

The paper is well structured and well written. The images nicely supports the narrative of the paper and provides a better understanding. The mix of insights from the biological domain area and the description of the tool and the decisions underlying the tool make it a nice reading experience.

Some of the section might be made more succinct such as the requirements which could be a little bit more memorable so its easier to follow them through the paper. Additionally, the two novel visualizations could have taken a little bit more space as I found it to be very short to catch the details and understand it correctly.

THe user evaluation start a slow. I would make these cases studies a little bit shorter and add some more analysis of the tool's usage. The web traffic logs analysis make a nice point of showing that the tool is being used not only by the research group it was designed for and therefore fills a gap. However, this could be said in one sentence in the summary to gain some space for describing the two novel plots.
I would also suggest to share the interview questions and anonymized responses as this would help other researchers doing work in the same or similar domain.

Making this paper great would be to add some details about how these two new visualizations were developed and validated. And adding for example some more details about the process of the collaboration such as how did the authors engage with the biologist, was it workshop based, were the authors embedded in the research groups in some way or another, and how did the authors manage the interests of the different research group.



- Wheat case studies (p.9) first word on the page, should it not be "... through ..."
- Figure is sometimes capitalized sometime not
- p5: Dot Plot: 2nd sentence should probably be "Dot plots .... "

---

### Meta-Review · Area_Chair1 · 2020-04-21

**Recommendation:** Accept
**Confidence:** 3

**Metareview:**

While the reviewers applauded for the topic and presentation of the paper, they brought some serious concerns. In particular, R1 and R2 thought that the system lacks novelty, and all reviewers found the evaluation can be stronger. R1 asked for fewer case studies but more depth. R2 requested for a more rigorous qualitative evaluation. R3 demanded shorter case studies and more analysis on the web traffic logs (R1 shared the same view).

Overall, while this paper is acceptable, but it needs "shepherding". Given this is the last deadline of GI, it might be not possible for allowing such a revision within the review cycle. But I keep a positive attitude.

---

### Decision · Program_Chairs · 2020-04-25

Accept